# Higher social tolerance is associated with more complex facial behavior in macaques

**Alan V Rincon[1]\*, Bridget M Waller[2], Julie Duboscq[3], Alexander Mielke[4], Claire Pérez[1], Peter R Clark[1,5], Jérôme Micheletta[1]**

[1]Department of Psychology, Centre for Comparative and Evolutionary Psychology, University of Portsmouth, Portsmouth, United Kingdom; [2]Centre for Interdisciplinary Research on Social Interaction, Department of Psychology, Nottingham Trent University, Nottingham, United Kingdom; [3]CNRS-MNHN-Université Paris Cité, Paris, France; [4]School of Biological and Behavioural Sciences, Queen Mary University of London, London, United Kingdom; [5]School of Psychology, University of Lincoln, Lincoln, United Kingdom

**\*For correspondence:**
avrincon1@gmail.com

**Competing interest:** The authors declare that no competing interests exist.

**Abstract** The social complexity hypothesis for communicative complexity posits that animal societies with more complex social systems require more complex communication systems. We tested the social complexity hypothesis on three macaque species that vary in their degree of social tolerance and complexity. We coded facial behavior in >3000 social interactions across three social contexts (aggressive, submissive, affiliative) in 389 animals, using the Facial Action Coding System for macaques (MaqFACS). We quantified communicative complexity using three measures of uncertainty: entropy, specificity, and prediction error. We found that the relative entropy of facial behavior was higher for the more tolerant crested macaques as compared to the less tolerant Barbary and rhesus macaques across all social contexts, indicating that crested macaques more frequently use a higher diversity of facial behavior. The context specificity of facial behavior was higher in rhesus as compared to Barbary and crested macaques, demonstrating that Barbary and crested macaques used facial behavior more flexibly across different social contexts. Finally, a random forest classifier predicted social context from facial behavior with highest accuracy for rhesus and lowest for crested, indicating there is higher uncertainty and complexity in the facial behavior of crested macaques. Overall, our results support the social complexity hypothesis.

## eLife assessment

This study shows **important** evidence of the correlation between social tolerance and communicative complexity in a comparison of three macaque species. Notably, the authors use an innovative, detailed methodology for quantifying facial expressions during social interactions. The results are **convincing** regarding a positive association between social complexity and facial behaviour, which should stimulate further comparative research in this field.

## Introduction

Animals must overcome a range of environmental and ecological challenges to survive and reproduce, with group-living species having to overcome additional social challenges to maximize fitness. Communicative signals can be used to navigate a number of different social situations and may need to become more elaborate as social complexity increases. The social complexity hypothesis for

communicative complexity encapsulates this idea, proposing that animal societies with more complex social systems require more complex communication systems (*Freeberg et al., 2012*).

The social complexity hypothesis has become a topical issue in recent years, with questions regarding the definitions, measurement, and selective pressures driving both social and communicative complexity (*Peckre et al., 2019*; *Raviv et al., 2022*). Social complexity as experienced by group members can be affected by the level of differentiation of social relationships, where complexity increases as social relationships become more differentiated (*Bergman and Beehner, 2015*; *Aureli et al., 2022*). In a socially complex society, individuals interact frequently with each other in diverse ways and in many different contexts (*Freeberg et al., 2012*). If the types of interactions that individuals have is constrained, for example, by dominance or kinship, then social complexity decreases (*Freeberg et al., 2012*). Social complexity is also affected by the predictability or consistency of social interactions (*Aureli et al., 2022*; *Aureli and Schino, 2019*). When the behavior of social partners is unpredictable, such as when the dominance hierarchy is unstable, individuals likely perceive the social environment as more complex (*Aureli and Schino, 2019*). These operational definitions of social complexity are valuable to advance the study of social complexity but are not easy to quantify with a single measure (*Kappeler, 2019*).

Similarly, communicative complexity is also difficult to quantify. Many studies have used the number of signaling units as a measure of communicative complexity (*Peckre et al., 2019*). While a useful measure, it is not always apparent what a signaling unit is. For example, calls are sometimes graded on a continuous scale without a clear separation between different call types (*Keenan et al., 2013*). Fewer studies have investigated the complexity of non-vocal communication (*Freeberg et al., 2012*; *Peckre et al., 2019*), but similar issues exist. One previous study quantified the repertoire of facial behavior in macaques by the number of discrete facial expressions that a species displays and found that it was positively correlated with conciliatory tendency and counter-aggression across species (*Dobson, 2012*). However, classifying facial expressions into discrete categories (e.g., bared-teeth display) does not capture the full range of expressiveness and meanings that the face can convey. For example, subtle morphological variations in bared-teeth displays are associated with different outcomes of social interactions (e.g., affiliation versus submission) in crested macaques (*Macaca nigra*) (*Clark et al., 2020*). A better approach is to quantify facial behavior at the level of individual facial muscle movements (*Waller et al., 2020*), which can be done using the Facial Action Coding System (FACS) (*Ekman et al., 2002*). In FACS, visible muscle contractions in the face are called Action Units and allow for a detailed and objective description of facial behavior (*Waller et al., 2020*; *Ekman et al., 2002*). Indeed, facial mobility, as defined by the number of Action Units that a species has, is positively correlated with group size across non-human primates (*Dobson, 2009a*). However, isolated muscle movements still do not account for the full diversity of facial behavior because facial muscles often contract simultaneously to produce a large variety of distinct facial expressions.

One promising avenue to approximate complexity in living organisms is to quantify the uncertainty or predictability of a system (*Rebout et al., 2021*; *Sambrook and Whiten, 1997*), which are general properties of complex systems (*McDaniel and Driebe, 2005*; *Schuster, 2016*). Shannon's information entropy (*Shannon, 1948*) is a measure of uncertainty that can be applied to animal communication. Conceptually, entropy measures the potential amount of information that a communication system holds, rather than what is actually communicated (*Shannon, 1948*; *Adami, 2002*). Entropy increases along two dimensions: (1) with increasing diversity of signals and (2) as the relative frequency of signal use becomes more balanced. For example, a system with three calls can hold more information than a system with one call and thus would have higher entropy. Likewise, a system with three calls used with equal frequency will have a higher entropy than another system that expresses one call more frequently than the two others. Uncertainty increases with entropy because each communicative event has the potential to derive from a greater number of units. The relative entropy, or uncertainty, of different systems can be compared by calculating the ratio between the observed and maximum entropy of each system.

The predictability and uncertainty of a communication system is also affected by how flexibly signals are used across different social contexts (*Aureli et al., 2022*). For instance, if signal A is always used in an aggressive context and signal B is always used in an affiliative context, then it is easy to predict the context from the signal. Conversely, if signals A and B are used in both contexts, then predictability is lower, and complexity is higher. Extremely rare signals do not substantially affect the predictability

of a system regardless of whether they have high or low specificity since they are seldom observed in the majority of social interactions. Therefore, predictability is highest when signals are both highly context-specific and occur in that context often. Additionally, predictability can be measured directly by training a machine learning classifier to predict the social context that a given signal was used in. Differences in prediction error would approximate the relative uncertainty and complexity, with accuracy being lower in more complex systems. However, as complexity lies somewhere between order and randomness (*Sambrook and Whiten, 1997*; *Adami, 2002*), we should still be able to predict the social contexts better than chance, even in a complex system.

Studying closely related species offers a robust means of testing the social complexity hypothesis due to their homologous communication systems. For this reason, macaques (genus *Macaca*) are excellent taxa to test the social complexity hypothesis. All species have a similar social organization consisting of multi-male, multi-female groups, but vary in social style in ways that are highly relevant to predictions of the social complexity hypothesis. The social styles of macaques consist of several covarying traits that can be ordered along a social tolerance scale ranging from the least (grade 1) to most tolerant (grade 4) (*Thierry, 2007*; *Thierry, 2022*). Social interactions for the least tolerant species, such as rhesus (*Macaca mulatta*) and Japanese (*Macaca fuscata*) macaques, are generally more constrained by a steep linear dominance hierarchy (*Balasubramaniam et al., 2012*) and nepotism (*Sueur et al., 2011*; *Thierry and Berman, 2010*; *Duboscq et al., 2013*). Additionally, severe agonistic interactions are more frequent (*Duboscq et al., 2013*), instances of counter-aggression and reconciliation after conflicts are rare (*Balasubramaniam et al., 2012*; *Duboscq et al., 2013*), and formal signals of submission are commonly used (*de Waal and Luttrell, 1985*; *Preuschoft and Schaik, 2000*). Combined, these behavioral traits indicate that agonistic interactions of the least tolerant species are more stereotyped and formalized. Thus, the outcome of such interactions is more certain, whereas the opposite is true for the most tolerant species, such as crested and Tonkean (*Macaca tonkeana*) macaques. The unpredictability in the outcome of agonistic interactions of tolerant macaques potentially results in a social environment that is perceived as more complex by individuals (*Aureli and Schino, 2019*), where more subtle means of negotiation during conflicts may be necessary.

In this study we compared the facial behavior of three macaque species that vary in their degree of social tolerance and, therefore, social complexity: rhesus (least tolerant), Barbary (*Macaca sylvanus*, mid-tolerant), and crested macaques (most tolerant). For macaques (and primates in general), the face is central to communication and is a key tool in allowing individuals to achieve their social goals by communicating motivations, emotions, and/or intentions (*Waller et al., 2017*; *Fridlund, 1994*). We coded facial behavior at the level of individual visible muscle movements using FACS and recorded all observed unique combinations, rather than classifying facial expressions into discrete categories. Based on the social complexity hypothesis (*Freeberg et al., 2012*), we expected that tolerant species would have higher communicative complexity, given that their social relationships are less constrained by dominance and have higher overall uncertainty in the outcome of agonistic interactions. Specifically, we predicted the following: (1) relative entropy of facial behavior will be lowest in the rhesus and highest in crested macaques, (2) context specificity of facial behavior will be highest in rhesus and lowest in crested macaques, and (3) social context can be predicted from facial behavior most accurately in rhesus and least accurately in crested macaques. For all three metrics, we expected Barbary macaques to lie somewhere in-between the rhesus and crested macaques.

## Results
### Entropy of facial behavior

To compare the relative uncertainty in the facial behavior of macaques, we defined facial behavior by the unique combination of Action Units (facial muscle movements) that occurred at the same time. We calculated the entropy ratio for each species and social context, defined as the ratio between the observed entropy and the expected entropy if Action Units were used randomly. Values closer to 0 indicate that there is low uncertainty (e.g., when only a few facial movements are used frequently) and values closer to 1 indicate high uncertainty (e.g., when many facial movements are used frequently). To determine whether the entropy ratios for each species differed within social context, we calculated the entropy ratio on 100 bootstrapped samples of the data, resulting in a distribution of possible values. The bootstrapped entropy ratio of facial behavior differed across species and within social

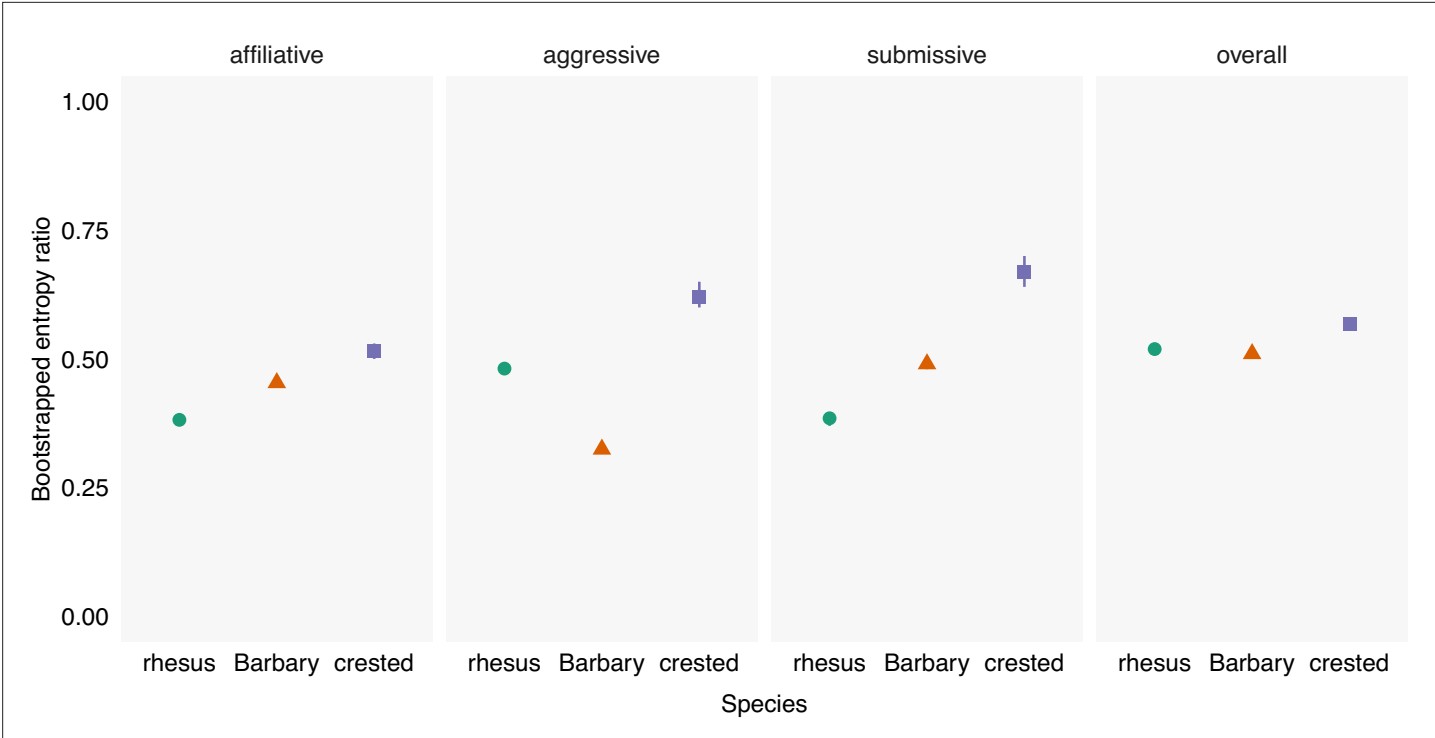

**Figure 1.** Bootstrapped entropy ratio of facial behavior across social contexts for three species of macaques. The entropy ratio was calculated on 100 bootstrapped samples of the data by dividing the observed entropy by the expected entropy if Action Units were used randomly for each social context. The entropy ratio ranges from 0 to 1, with higher values indicating higher uncertainty. Symbols and whiskers indicate mean and range of bootstrapped values.

contexts (*Figure 1*). In an affiliative context, the entropy ratio was highest for crested, then Barbary, and lowest for rhesus macaques (crested: mean = 0.52, range = 0.50–0.53; Barbary: mean = 0.45, range = 0.45–0.46; rhesus: mean = 0.38, range = 0.37–0.39). In an aggressive context, the entropy ratio was highest for crested, then rhesus and lowest for Barbary macaques (crested: mean = 0.62, range = 0.60–0.65; Barbary: mean = 0.32, range = 0.32–0.33; rhesus: mean = 0.48, range = 0.47–0.49). In a submissive context, the entropy ratio was highest for crested, then Barbary, and lowest for rhesus macaques (crested: mean = 0.67, range = 0.64–0.70; Barbary: mean = 0.49, range = 0.48–0.50; rhesus: mean = 0.38, range = 0.37–0.39). Overall, across all contexts, including when the context was unclear, the entropy ratio was highest for crested, and similar for Barbary and rhesus macaques (crested: mean = 0.57, range = 0.56–0.58; Barbary: mean = 0.51, range = 0.51–0.51; rhesus: mean = 0.52, range = 0.51–0.52; *Figure 1*).

## Context specificity of facial behavior

We calculated the context specificity for all possible combinations of Action Units. Here, we report specificity for combinations that were observed in at least 1% of observations per species and social context because extremely rare signals do not affect the predictability of a system substantially, regardless of whether they have high or low specificity. Specificity for each Action Unit combination was defined as the number of times it was observed in one context divided by the total number of times it was observed across all contexts. When considering single Action Units, some were observed in only one context, but most were observed at least once in all three contexts for all three species (*Figure 2*). On average, single Action Units were observed in fewer contexts for rhesus (mean degree = 1.9), compared to Barbary (mean degree = 2.4), and crested macaques (mean degree = 2.6). The specificity of all Action Unit combinations used in an affiliative context was highest for the rhesus macaques, then Barbary, and lowest for crested macaques (rhesus: mean = 0.80, SD = 0.28, n=69; Barbary: mean = 0.63, SD = 0.26, n=450; crested: mean = 0.37, SD = 0.26, n=327; *Figure 3a*). The specificity of Action Unit combinations used in an aggressive context was highest for rhesus, then

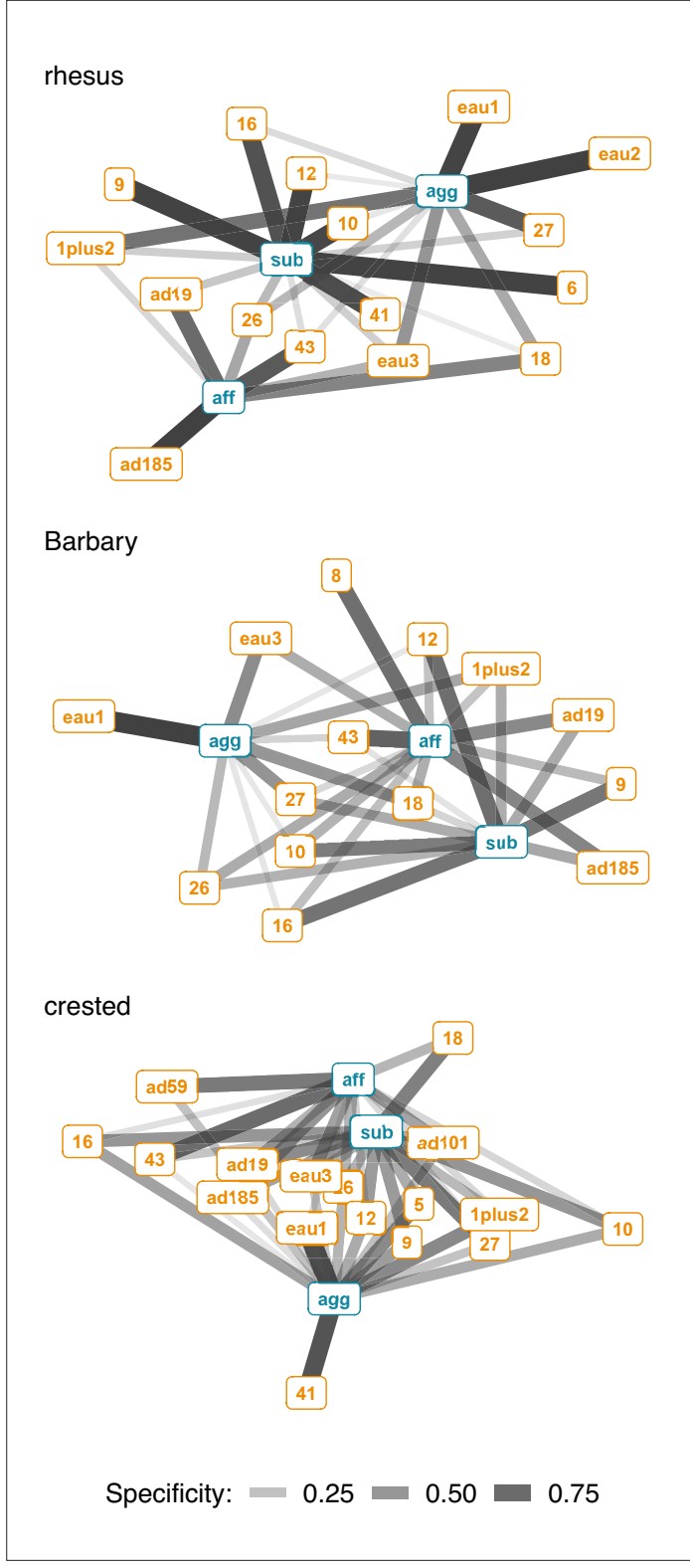

**Figure 2.** Bipartite network of single Action Units (orange) and social context (blue) for three species of macaques. Edges are shown for Action Units that occurred in at least 1% of observations per context. Edge thickness and transparency are weighted by specificity, which ranges from 0 (indicating an Action Unit is never observed in a context) to 1 (indicating an Action Unit is only observed in one context). Context abbreviations: agg = aggressive, aff = affiliative, sub = submissive.

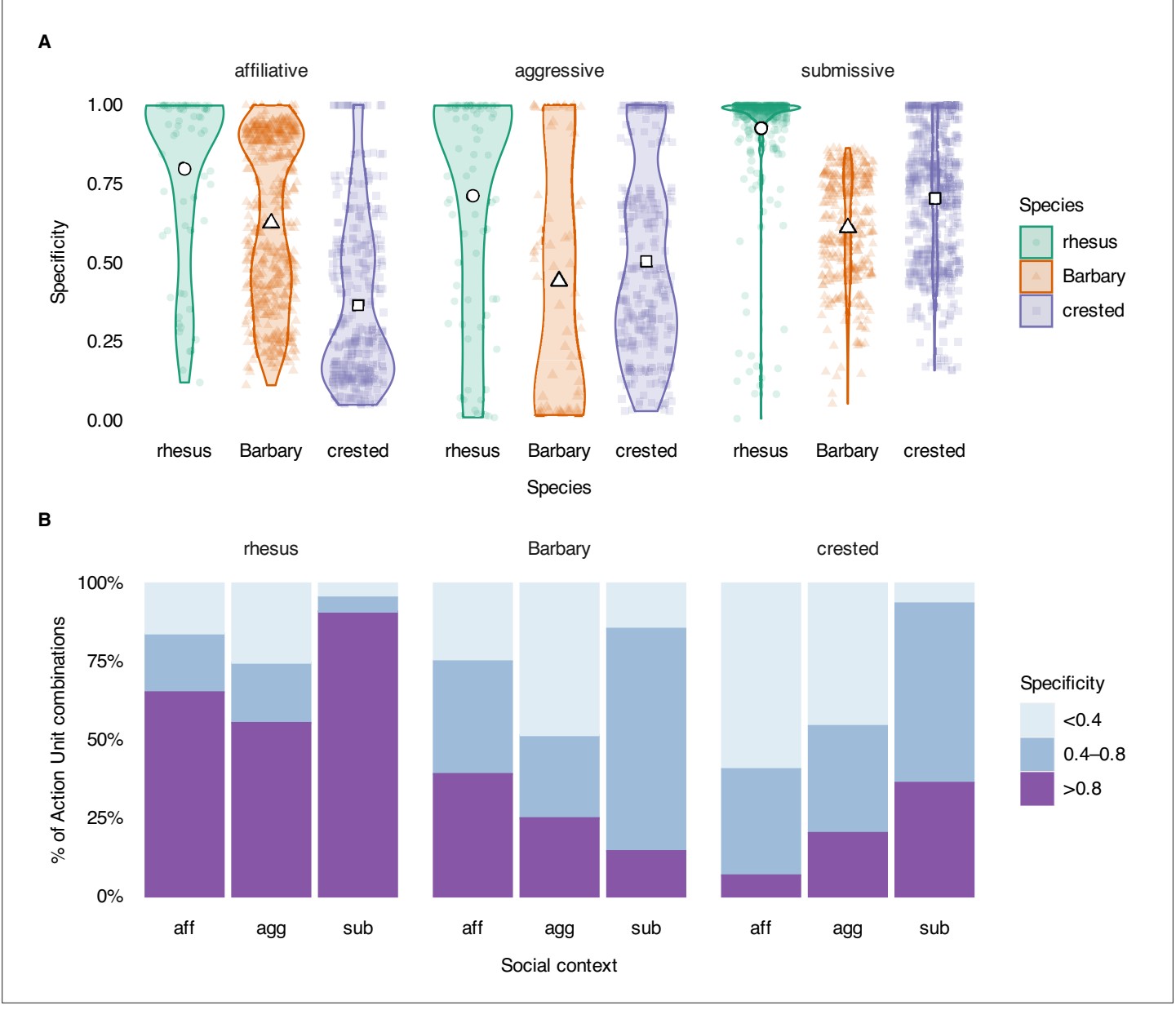

**Figure 3.** Specificity of Action Unit combinations that were used in at least 1% of observations per species per social context. Specificity ranges from 0 (indicating an Action Unit is never observed in a context) to 1 (indicating an Action Unit is only observed in one context). (**A**) Distribution of Action Unit combination specificity. Width of violin plots indicate the relative density of the data. Colored symbols indicate unique Action Unit combinations. White symbols indicate mean specificity. (**B**) Proportion of Action Unit combinations used with high (>0.8), moderate (0.4–0.8), or low (<0.4) specificity. Context abbreviations: agg = aggressive, aff = affiliative, sub = submissive.

crested, and lowest for Barbary macaques (rhesus: mean = 0.71, SD = 0.35, n=83; Barbary: mean = 0.44, SD = 0.38, n=64; crested: mean = 0.51, SD = 0.30, n=281). The specificity of Action Unit combinations used in a submissive context was also highest for rhesus, then crested, and lowest for Barbary macaques (rhesus: mean = 0.93, SD = 0.18, n=312; Barbary: mean = 0.61, SD = 0.18, n=297; crested: mean = 0.70, SD = 0.21, n=595). The majority (>50%) of Action Unit combinations used by rhesus macaques had high specificity (>0.8) in all three social contexts, whereas only a minority (<50%) of Action Unit combinations used by Barbary and crested macaques had high specificity (*Figure 3b*).

**Table 1.** Confusion matrices for random forest classifier predictions of social context from Action Unit combinations.

| | Truth | | |
| Prediction | Affiliative | Aggressive | Submissive |
| --- | --- | --- | --- |
| Rhesus | | | |
| Affiliative | 636 | 19 | 9 |
| Aggressive | 81 | 1205 | 17 |
| Submissive | 2 | 6 | 731 |
| Barbary | | | |
| Affiliative | 2573 | 24 | 442 |
| Aggressive | 200 | 1219 | 165 |
| Submissive | 166 | 34 | 528 |
| Crested | | | |
| Affiliative | 1134 | 90 | 43 |
| Aggressive | 16 | 86 | 11 |
| Submissive | 3 | 1 | 7 |

## Predicting social context from facial behavior

A random forest classifier was able to predict social context (affiliative, aggressive, or submissive) from facial behavior with a better accuracy than expected by chance alone for all three species of macaques. The classifier was most accurate for rhesus (kappa = 0.92), then Barbary (kappa = 0.68), and least accurate for crested macaques (kappa = 0.49). The confusion matrices for model predictions are shown in *Table 1*.

## Discussion

We investigated the hypothesis that complex societies require more complex communication systems (*Freeberg et al., 2012*) by comparing the complexity of facial behavior of three species of macaques that vary in their degree of social tolerance and complexity. We defined facial behavior by the unique combinations of muscle movements visible in the face. Doing so allows for a much more precise description of facial behavior and captures subtle differences that are lost if facial expressions are classified as discrete categories. We quantified communicative complexity using three measures of uncertainty and predictability: entropy, context specificity, and prediction error. Collectively, our results suggest that the complexity of facial behavior is higher in species with a more tolerant—and therefore more complex—social style; complexity was highest for crested, followed by Barbary, and lowest in rhesus macaques. In light of what we know about the differences between macaque social systems, our results support the predictions of the social complexity hypothesis for communicative complexity.

The entropy ratio of facial behavior was highest in crested compared to Barbary and rhesus macaques, both overall and within each social context (affiliative, aggressive, submissive). This result suggests that crested macaques use a higher diversity of facial signals within each social context more frequently, resulting in the higher relative uncertainty in their use of facial behavior. Information theory defines information as the reduction in uncertainty once an outcome is learned (*Shannon, 1948*). By this definition, our data suggest that the facial behavior of crested macaques has the *potential* to communicate more information, compared to Barbary and rhesus macaques, although this would need to be explicitly tested in future studies. Our findings are in line with predictions of the social complexity hypothesis (*Freeberg et al., 2012*) given the differences in social styles between tolerant and intolerant macaques. In tolerant macaque societies, social interactions are less constrained by dominance (*Balasubramaniam et al., 2012*) such that rates of counter-aggression and reconciliation post-conflict are higher (*Duboscq et al., 2013*; *Thierry et al., 2008*). Thus, there is a greater variability in the kind of interactions that individuals have, potentially requiring the use of more diverse facial

behavior to achieve social goals, particularly during conflicts. Similarly, strongly bonded chimpanzee (*Pan troglodytes*) dyads exhibit a larger repertoire of gestural communication than non-bonded dyads, presumably due to the former having more varied types of social interactions (*Amici and Liebal, 2022*).

The overall entropy ratio of rhesus and Barbary macaques was similar, suggesting that they have similar communicative capacity using facial behavior. However, the entropy ratio differed when compared within social contexts; while relative entropy was higher for Barbary macaques in affiliative and submissive contexts, it was higher for rhesus macaques in aggressive contexts. One possible explanation may be due to the use of stereotyped signals of submission and dominance in each species. For example, subordinate rhesus macaques regularly exhibit stereotyped signals of submission (silent-bared-teeth), whereas dominant Barbary macaques regularly exhibit stereotyped threats (round-open-mouth) (*de Waal and Luttrell, 1985*; *Preuschoft and Schaik, 2000*). Frequent use of a stereotyped signal within a context reduces the overall diversity of signals, resulting in a lower entropy ratio for submission and aggression in rhesus and Barbary macaques, respectively. It has been suggested that in societies with high power asymmetries between individuals, such as in rhesus macaques, spontaneous signals of submission serve to prevent conflicts from escalating as well as increasing the tolerance of dominant individuals toward subordinates (*Preuschoft and Schaik, 2000*). In societies with more moderate power asymmetries, such as in Barbary macaques, subordinates may be less motivated to spontaneously submit and thus dominants may need to assert their dominance with formalized threats more frequently (*Preuschoft and Schaik, 2000*).

While the entropy ratio captures the uncertainty of facial behavior used within a social context, context specificity captures the uncertainty generated when the same facial behavior is used flexibly across different social contexts. Overall, the context specificity of facial behavior was higher for the intolerant rhesus macaques as compared to the more tolerant Barbary and crested macaques across all three social contexts. This pattern occurred for both the mean specificity values and the proportion of Action Unit combinations used that had high (>0.8) specificity. Similarly, a previous study demonstrated that vocal calls of tolerant macaques are less context specific than in intolerant macaques (*Rebout et al., 2022*). There was not a clear difference in specificity between Barbary and crested macaques; specificity was higher for Barbary macaques in affiliative contexts, similar for both species in aggressive contexts, and higher for crested macaques in submissive contexts. These differences in context specificity of communicative signals across macaque species may be related to differences in power asymmetry in their respective societies, particularly as it relates to the risk of injury. For macaques, bites are far more likely to injure opponents than other types of contact aggression (e.g., grab, slap) and thus provide the best proxy for risk of injury (*Thierry, 2022*). The percentage of conflicts involving bites is much higher in the less tolerant rhesus macaque, compared to the more tolerant Barbary and crested macaques who have similar low rates of aggression involving bites (*Duboscq et al., 2013*; *Tyrrell et al., 2020*). Risky situations may promote the evolution of more conspicuous, stereotypical signals to reduce ambiguity (*Clark et al., 2022*). Indeed, intolerant macaques such as the rhesus more commonly use formal signals of submission (*de Waal and Luttrell, 1985*; *Preuschoft and Schaik, 2000*). In our study, rhesus macaques used facial behavior with high specificity across all contexts but particularly in submissive contexts. If the same facial behavior (or signal in general) is used in multiple social contexts, its meaning may be uncertain and must be deduced from additional contextual cues (*Seyfarth and Cheney, 2017*). When facial behavior is highly context specific, there is less uncertainty about the meaning of the signal and/or intention of the signaler. In a society where the risk of injury from aggression is high, it may be adaptive for individuals to use signals that are highly context specific or ritualized to reduce uncertainty about its meaning. By contrast, the lower risk of injury in Barbary and crested macaques may allow room for a greater variety of more nuanced behaviors during conflicts as well as higher rates of reconciliation post-conflict (*Duboscq et al., 2013*; *Thierry et al., 2008*).

In all three species of macaques, at least some facial muscle movements had low specificity and were therefore used across multiple social contexts that likely differed in valence. This finding is in line with the idea that communicative signals in primates are better interpreted as the signaler announcing its intentions and likely future behavior (*Cheney and Seyfarth, 2018*; *Fischer and Price, 2017*), and not necessarily as an expression of emotional state (*Waller et al., 2017*; *Fridlund, 1994*; *Cheney and Seyfarth, 2018*; *Barrett et al., 2019*).

We found that a random forest classifier was least accurate at predicting social context from facial behavior for crested, followed by Barbary, and then rhesus macaques. The behavior of complex systems is generally harder to predict than simpler ones (*McDaniel and Driebe, 2005*; *Schuster, 2016*). Thus, the relatively poorer performance of the classifier in crested macaques suggests that they have the most complex facial behavior. Nevertheless, the classifier was able to predict social context from facial behavior with better accuracy than expected by chance alone for all three species of macaque, including the crested. This result confirms the assumption that facial behavior in macaques is not used randomly and most likely has some communicative or predictive value (*Waller et al., 2016*). It is worthwhile to reiterate here that completely random (and thus unpredictable) systems are not considered complex (*Adami, 2002*). Therefore, the species with the highest entropy values, or unpredictability, could be interpreted as having a simpler communication system than a species with a moderately high entropy value or unpredictability. But the communications systems of living organisms are unlikely to be observed as random, otherwise they would not have evolved as signals. Therefore, working under the assumption that animal communication systems cannot possibly be random, we can conclude that the species whose communication system has the highest relative entropy and unpredictability is in fact the most complex (*Rebout et al., 2021*).

In addition to social complexity, it is possible that other factors are related to the complexity of facial behavior. For example, primates with a larger body size have greater facial mobility (*Dobson, 2009a*; *Santana et al., 2014*), which could allow for greater complexity of facial behavior. However, differences in mean body mass across the three macaques species of this study are small (rhesus: 6.5 kg; Barbary: 11.5 kg; crested: 7.4 kg) (*Jones et al., 2009*) with substantial overlap in body weight across adult individuals of the different species (*Smith and Jungers, 1997*), and so it is unlikely to explain the differences in the complexity of facial behavior that we report in this study. The degree of terrestriality could also influence the evolution of facial signals due to more limited visibility in the canopy. However, differences in facial mobility across terrestrial and non-terrestrial primates are not significant once body size is controlled for (*Dobson, 2009a*). Furthermore, all three species included in this study have comparable levels of terrestriality, spending the majority (52–72%) of the time on the ground (*Khatiwada et al., 2020*; *O'Brien and Kinnaird, 1997*; *El Alami and Chait, 2014*). Spatial spread is another factor that could influence the use of facial signals. For example, when group spread is higher, reliance on facial signals could be lower since it is harder to perceive facial signals from a large distance. There are currently no reliable data on spatial spread of the three species of this study in their natural habitat but it could be a good avenue for future studies. It is also important to note that our study is correlational in nature and we cannot determine the direction of the link between social and communicative complexity. It is possible that an increase in communicative complexity evolved first, which then allowed for the evolution of more complex social systems. Finally, effectively, our comparison is limited to three species which is a small sample. However, the methodology we used is applicable to any species for which FACS is available (including other non-human primates, dogs, and horses; *Waller et al., 2020*), and therefore, we hope that other datasets will complement ours in the future.

Our results on the complexity of facial behavior in macaques is mirrored by previous studies showing that the complexity of vocal calls is similarly higher in tolerant compared to intolerant macaques (*Rebout et al., 2022*; *Rebout et al., 2020*). Although not all macaque facial expressions have a vocal component, vocalizations are fundamentally multisensory with both auditory and visual components, where different facial muscle contractions are partly responsible for different-sounding vocalizations (*Ghazanfar and Takahashi, 2014*). Indeed, some areas of the brain in primates integrate visual and auditory information resulting in behavioral benefits (*Ghazanfar and Eliades, 2014*). For example, macaques detect vocalizations in a noisy environment faster when mouth movements are also visible, where faster reaction times are associated with a reduced latency in auditory cortical spiking activity (*Chandrasekaran et al., 2013*). Combined, these findings suggest that the evolution in the complexity of vocal and facial signals in macaques may be linked and the same may be true of primates in general. For instance, humans not only have the most complex calls (language) and gestures, but most likely use the most complex facial behavior as well, given that their general facial mobility is highest among primates (most Action Units) (*Ekman et al., 2002*; *Dobson, 2009b*). In lemurs (Lemuriformes), the repertoire size of vocal, visual, and olfactory signals positively correlate with group size and each other, suggesting that complexity in all three communicative modalities

coevolved with social complexity (*Fichtel and Kappeler, 2022*). While the complexity of different communication modalities is likely interlinked and correlated with each other, future studies would ideally integrate signals from all modalities into a single communicative repertoire for each species. While collecting and analyzing data on multiple modalities of communication has historically been a challenge, such endeavors would be an important next step in the study of animal communication (*Liebal et al., 2022*). By breaking down signaling units to their smallest components, as we have done for facial behavior in this study, we may be able to define a 'signal' by temporal co-activation of visual, auditory, and perhaps even olfactory cues, which would provide the most comprehensive picture of animal communication.

## Methods

### Study subjects and data collection

Behavioral data and video recordings were collected on one adult male and 31 adult female rhesus macaques (*M. mulatta*), on 18 adult male and 28 adult female Barbary macaques (*M. sylvanus*), and 17 adult male and 21 adult female crested macaques (*M. nigra*). Admittedly, a more balanced sample size per sex would have been preferable for rhesus macaques. Nevertheless, male and female macaques must (and do) interact and communicate with each other regularly. Therefore, we have no a priori reason to expect an overall difference in the diversity and complexity of facial behavior between the sexes. The social complexity hypothesis makes predictions at the level of societies, and we feel like our sample size for rhesus macaques is large enough to representatively capture the complexity of their facial behavior.

Rhesus macaques belonged to one breeding group (Gruppe 1) at the German Primate Center, Germany. Monkeys were housed in naturalistic outdoor enclosure (approximately 290 m² and 4–7 m high) with free access to a heated indoor area (approximately 80 m² and 5–7 m high), which were enriched with ropes, logs, swings, and a small pond. Monkeys were fed daily a variety of fruits and vegetables, nuts, seeds, cereals, commercial monkey pellets, and had ad libitum access to water. All observations, including the recording of videos, were conducted outside of the enclosures. Data collection on the rhesus macaques took place between June and October 2021. Barbary macaques belonged to one group (German Group) out of two groups living at Trentham Monkey Forest, UK. Monkeys were able to freely move within a 24-hectare open enclosure of forest and grassy areas. Monkeys were fed daily a variety of fruits, vegetables, seeds, and monkey chow, and had ad libitum access to water. Data collection on the Barbary macaques took place between August and November 2019. Crested macaques belonged to two wild groups (R2A and PB1B) living in Tangkoko-Batuangus Nature reserve, North Sulawesi, Indonesia, and observed within the Macaca Nigra Project (http://www.macaca-nigra.org). Monkeys were not provisioned by humans and fed on natural foods and were habituated to the presence of human observers. Data collection on the crested macaques took place between December 2018 and April 2019.

For all study groups and subjects, focal animal observations (*Altmann, 1974*) lasting 15–30 min were conducted throughout the day in a pseudo-randomized order such that the number of days and time of day that each individual was observed was balanced. Videos of social interactions were recorded with a recording camera (Panasonic HDC-SD700, Bracknell, UK) during focal animal observations as well as ad libitum. Social behavior, including grooming, body contact, and agonistic interactions, was recorded using a handheld smartphone or tablet with purpose-built software (rhesus: Animal Behavior Pro [*Newton-Fisher, 2020*]; Barbary: CyberTracker [http://cybertracker.org], crested: Microsoft Excel).

### Facial behavior and social context coding

Facial behavior was coded at the level of observable individual muscle movements using the FACS (*Ekman et al., 2002*), adapted for each species of macaque (MaqFACS): rhesus (*Parr et al., 2010*), Barbary (*Julle-Danière et al., 2015*), crested (*Clark et al., 2020*). In FACS, individual observable muscle contractions are coded as unique Action Units (AUs; e.g., upper lip raiser AU10). Some common facial movements where the underlying muscle is unknown are coded as Action Descriptors (ADs; e.g., jaw thrust AD29). In MaqFACS, the lip-pucker AU18 has two subtle variations normally denoted as AU18i and AU18ii (*Parr et al., 2010*; *Julle-Danière et al., 2015*). However, it was often difficult to reliably

distinguish between these two subtle variations when coding videos, and so the lip-pucker was simply coded as AU18. We added a new Action Descriptor 185 (AD185) called jaw-oscillation, to denote the stereotyped movement of the jaw up and down. When combined with existing Action Units of lip movements, the jaw-oscillation AD185 allows for a more detailed and accurate coding of some facial behaviors that would otherwise be labeled as lipsmack (AD181), teeth-chatter, or jaw-wobble (*Clark et al., 2020*; *Parr et al., 2010*). A complete list of Action Units and Action Descriptors coded in this study is given in *Supplementary file 1—Table 1*.

We coded facial behavior of adult individuals but included their interactions with any other group member regardless of age or sex. Each social interaction was labeled with a context; aggressive, submissive, affiliative, or unclear. We did not consider interactions in a sexual context because data for the rhesus macaques were only collected during the non-mating season. Social context was labeled from the point of view of the signaler based on their general behavior and body language (but not the facial behavior itself), during or immediately following the facial behavior. An aggressive context was considered when the signaler lunged or leaned forward with the body or head, charged, chased, or physically hit the interaction partner. A submissive context was considered when the signaler leaned back with the body or head, moved away, or fled from the interaction partner. An affiliative context was considered when the signaler approached another individual without aggression (as defined previously) and remained in proximity, in relaxed body contact, or groomed either during or immediately after the facial behavior. In cases where the behavior of the signaler did not match our context definitions, or displayed behaviors belonging to multiple contexts, we labeled the social context as unclear. Social context was determined from the video itself and/or from the matching focal behavioral data, if available. Videos were FACS coded frame-by-frame using the software BORIS (*Friard et al., 2016*) by AVR (rhesus, Barbary, crested), CP (Barbary), and PRC (crested), who are certified FACS and MaqFACS coders. Inter-observer reliability was determined with the same index of agreement used by *Ekman et al., 2002*, for FACS, with the formula:

$$\frac{2\,(\text{The number of AUs on which both coders agreed})}{\text{Total number of AUs scored by both coders}}$$

An agreement rating of >0.7 was considered good (*Ekman et al., 2002*) and was necessary for obtaining certification. To obtain a MaqFACS coding certification, AVR, CP, and PRC coded 23 video clips of rhesus macaques and the MaqFACS codes were compared to the data of other certified coders (https://animalfacs.com). The mean agreement ratings obtained were 0.85, 0.73, 0.83 for AVR,

**Table 2.** Total number of social interactions per species and social context that were MaqFACS coded.

Note that combination of Action Units were grouped by time blocks of 500 ms. Therefore, the number of observations in the data is twice the duration of the social interaction in seconds.

| Species | Context | N interactions | N subjects | Duration (s) |
|---|---|---|---|---|
| | Affiliative | 193 | 29 | 1197 |
| | Aggressive | 413 | 32 | 2050 |
| | Submissive | 318 | 31 | 1262 |
| Rhesus | Unclear | 121 | 30 | 802 |
| | Affiliative | 683 | 43 | 4897 |
| | Aggressive | 585 | 44 | 2128 |
| | Submissive | 529 | 34 | 1890 |
| Barbary | Unclear | 603 | 45 | 3500 |
| | Affiliative | 241 | 35 | 1918 |
| | Aggressive | 62 | 23 | 284 |
| | Submissive | 25 | 18 | 115 |
| Crested | Unclear | 107 | 25 | 684 |

CP, and PRC, respectively. In addition, AVR and CP coded seven videos of Barbary macaques with a mean agreement rating of 0.79. AVR and PRC coded 10 videos of crested macaques with a mean agreement rating of 0.74.

*Table 2* shows the number of social interactions per species and context from which FACS codes were made.

### Statistical analyses

Prior to analyses, MaqFACS data were formatted as a binary matrix with Action Units and Action Descriptors (hereafter simply Action Units) in the columns. Each row denoted an observation time block of 500 ms, where if an Action Unit was active during this time block, it was coded 1 and coded 0 if not. Thus, each row contained information on the combination of facial muscle movements that were co-activated within a 500 ms time window (*Table 2*). All 500 ms time blocks per interaction were used in the statistical analyses in order to retain all the variation and complexity of the facial behavior (Action Unit combinations) used by the macaques. All statistical analyses were conducted in R (version 4.2.1) (*R Development Core Team, 2022*).

The observed entropy for each social context was calculated using Shannon's information entropy formula (*Shannon, 1948*):

$$H = -\sum_{i}^{n} p_i \log p_i$$

where *n* is the number of unique Action Unit combinations and p is the probability of observing each Action Unit combination in each social context. The expected maximum entropy was calculated by randomizing the data matrix while keeping the number of active Action Units per observation (row) the same. This process was repeated 100 times and the mean of the randomized entropy values was used as the expected entropy. Therefore, the expected entropy indicated the entropy of the system if

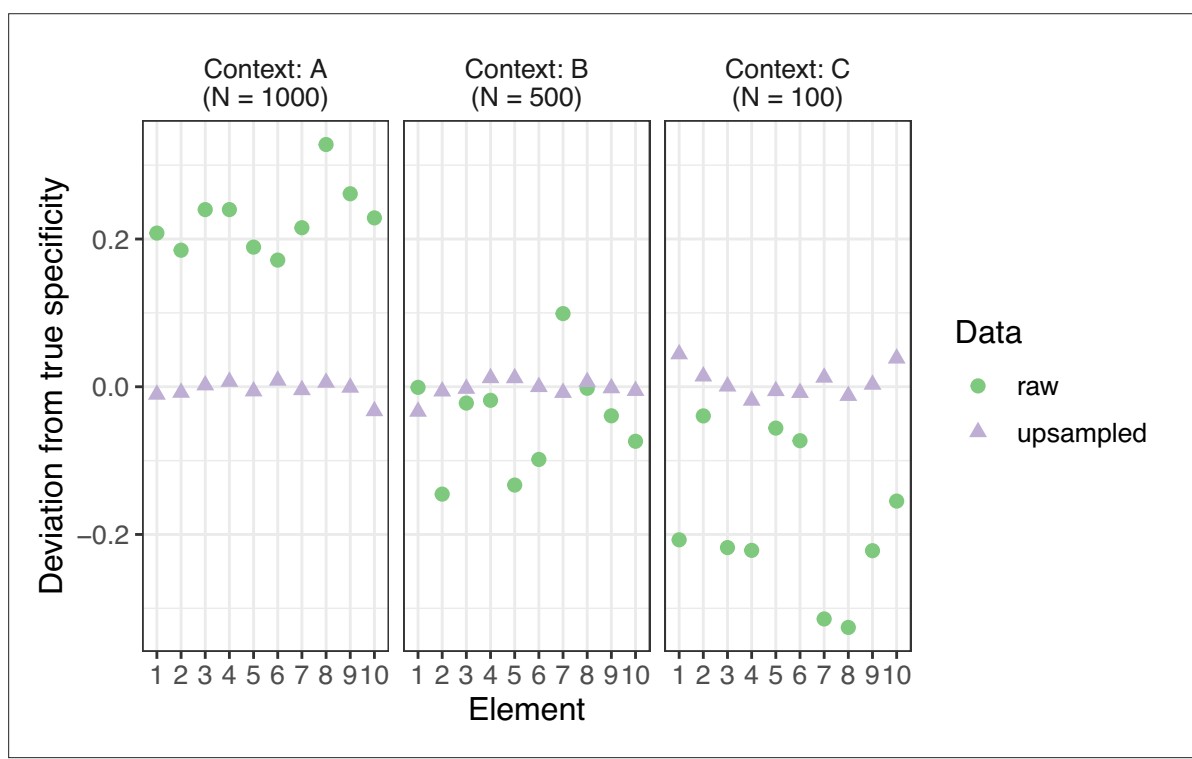

**Figure 4.** Calculating context specificity on an imbalanced dataset. Specificity was calculated on a simulated dataset with an imbalanced number of observations per context. The calculated specificity values deviated from the true specificity such that they were higher in the context with most observations and lower in the context with fewest observations (green circles). Randomly upsampling observations from the minority contexts (**B and C**) such that they have the same number of observations as the majority context (**A**) prior to calculating specificity minimized the bias in the calculated specificity values (purple triangles).

facial muscle contractions occurred at random, while keeping the combination size of co-active muscle movements within the range observed in the data. The entropy ratio was calculated by dividing the observed entropy by the expected (maximum) entropy. To determine whether the entropy ratios for each species differed within social context, the entropy ratio was calculated on 100 bootstrapped samples of the data, resulting in a distribution of possible entropy ratios. If the distribution of bootstrapped entropy ratios did not overlap, the differences between entropy ratios were considered to be meaningful.

We calculated the specificity with which Action Unit combinations are associated with a social context within each species using the function 'specificity' from the R package 'NetFACS' (version 0.5.0) (*Mielke et al., 2022*). Due to an imbalanced number of observations across social contexts, contexts with fewer observations were randomly upsampled prior to the specificity calculation. During the upsampling procedure, all observations of the minority contexts were kept, and new observations were randomly sampled to match the number of observations in the majority context. This procedure corrects for any bias in the specificity results from an imbalanced dataset (see Specificity bias correction section below for details; *Figure 4*). Specificity is the conditional probability of a social context given that an Action Unit combination is observed, and ranges from 0 (when an Action Unit combination is never observed in a context) to 1 (when an Action Unit is only observed in one context). Low specificity values indicate that Action Units were used flexibly across multiple contexts whereas high values indicate that Action Units were used primarily in a single context. Specificity was calculated for all Action Unit combination sizes ranging from 1 to 11 (the maximum observed combination size) co-active Action Units. When reporting context specificity results, we excluded Action Unit combinations that occurred in less than 1% of observations within a social context because extremely rare signals do not impact the predictability of a communication system regardless of whether specificity is low or high. Therefore, excluding rare Action Unit combinations removes noise from the specificity results. We report the mean specificity of Action Unit combinations per social context and the proportion of Action Unit combinations that have high, moderate, or low specificity. For single Action Units we plotted bipartite networks that show how Action Units are connected to social context weighted by their specificity.

To predict social context from the combination of Action Units we fit a random forest classifier using the 'tidymodels' R package (version 1.0.0) (*Kuhn and Wickham, 2020*) using the function 'ran_forest' with the engine set to 'ranger' (*Wright and Ziegler, 2017*), 500 trees, 4 predictor columns randomly sampled at each split, and 10 as the minimum number of data points in a node required for splitting further. The data were randomly split into a training set (70%) and a test set (30%), while keeping the proportion of observations per social context the same in the training and test sets. Due to an imbalanced number of observations across social contexts, contexts with fewer observations were over-sampled in the training set using the SMOTE algorithm (*Chawla et al., 2002*) to improve the classifier predictions. To assess the classifier performance, we report the kappa statistic, which denotes the observed accuracy corrected for the expected accuracy (*Cohen, 1960*). Kappa is 0 when the classifier performs at chance level and 1 when it shows perfect classification. Kappa values between 0 and 1 indicate how much better the classifier performed than chance (e.g., kappa of 0.5 indicates the classifier was 50% better than chance). Kappa is a more reliable estimate of model performance than accuracy alone when the relative sample size for each context is imbalanced, as was the case with our data.

## Specificity bias correction

FACS data were simulated for three contexts (A, B, C) and 10 elements (1–10, representing Action Units). Specificity was calculated when all contexts had an equal number of observations (denoting the true specificity) and on a subset of the data where the number of observations between the three contexts was imbalanced at a ratio of 10:5:1. Specificity values were skewed higher in the context with most observations (A) and skewed lower in context with fewest observations (C). Upsampling the minority contexts, such that all contexts had the same number of observations, substantially minimized the error bias in specificity values (*Figure 4*). The R script for the simulation can be found at https://github.com/avrincon/macaque-facial-complexity; copy archived at *Rincon, 2022*.

## Ethics

This work adhered to the Guidelines for the treatment of animals in behavioral research and teaching (*ASAB Ethical Committee and ABS Animal Care Committee, 2022*) and was approved by the Animal Welfare and Ethical Review Body of the University of Portsmouth (AWERB, approval number: 919B). The AWERB uses UK Home Office guidelines on the Animals (Scientific Procedures) Act 1986 when assessing proposals and adheres to the regulations of the European Directive 2010/63/EU. The German Primate Center also complies with the European Directive 2010/63/EU, as well as with the provisions of the German Animal Welfare Act.

## Acknowledgements

We thank the German Primate Center (DPZ) for permission to collect data on the rhesus macaques, Uwe Schönmann for logistical support, and Julia Ostner for being our host at the DPZ. We thank Matt Lowatt and Ellen Merz for permission to collect data on the Barbary macaques at Trentham Monkey Forest. We thank the Indonesian State Ministry of Research and Technology (RISTEK), the Directorate General of Forest Protection and Nature Conservation (PHKA) and the Department for the Conservation of Natural Resources (BKSDA), North Sulawesi, for permission to access groups of crested macaques in the Tangkoko-Batuangus Nature Reserve. We thank Christof Neumann for statistical advice. This work was funded by the Leverhulme Trust (RPG2018-334).

## Additional information

### Funding

| Funder | Grant reference number | Author |
| --- | --- | --- |
| Leverhulme Trust | RPG2018-334 | Jérôme Micheletta |

The funders had no role in study design, data collection and interpretation, or the decision to submit the work for publication.

### Author contributions

Alan V Rincon, Conceptualization, Data curation, Software, Formal analysis, Validation, Investigation, Visualization, Methodology, Writing - original draft, Writing - review and editing; Bridget M Waller, Julie Duboscq, Jérôme Micheletta, Conceptualization, Supervision, Funding acquisition, Methodology, Project administration, Writing - review and editing; Alexander Mielke, Software, Methodology, Writing - review and editing; Claire Pérez, Peter R Clark, Data curation, Investigation, Writing - review and editing

### Author ORCIDs

Alan V Rincon http://orcid.org/0000-0001-6181-0152
Bridget M Waller http://orcid.org/0000-0001-6303-7458
Jérôme Micheletta http://orcid.org/0000-0002-4480-6781

### Ethics

This work adhered to the Guidelines for the treatment of animals in behavioral research and teaching and was approved by the Animal Welfare and Ethical Review Body of the University of Portsmouth (AWERB, approval number: 919B). The AWERB uses UK Home Office guidelines on the Animals (Scientific Procedures) Act 1986 when assessing proposals and adheres to the regulations of the European Directive 2010/63/EU. The German Primate Center also complies with the European Directive 2010/63/EU, as well as with the provisions of the German Animal Welfare Act.

Reviewer #1 (Public Review): https://doi.org/10.7554/eLife.87008.3.sa1
Reviewer #2 (Public Review): https://doi.org/10.7554/eLife.87008.3.sa2
Author Response https://doi.org/10.7554/eLife.87008.3.sa3

## Additional files

### Supplementary files
• Supplementary file 1. A complete list of Action Units and Action Descriptors coded in this study.
• MDAR checklist

### Data availability
The data generated and analyzed in this study, along with the R code used for all statistical analysis is available on GitHub, https://github.com/avrincon/macaque-facial-complexity (copy archived at *Rincon, 2022*).

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
