## [Editor Report · eLife assessment]

This study shows **important** evidence of the correlation between social tolerance and communicative complexity in a comparison of three macaque species. Notably, the authors use an innovative, detailed methodology for quantifying facial expressions during social interactions. The results are **convincing** regarding a positive association between social complexity and facial behaviour, which should stimulate further comparative research in this field.

---

## [Referee Report · Reviewer #1 (Public Review)]

After revision, the manuscript is clearly improved and I thank the authors for their efforts. Yet, two contentious issues remain.

Firstly, I am skeptical whether the circularity issue has been resolved.

The authors equate uncertainty in the outcome of interactions with social complexity and they then diagnose for these three species that higher social complexity correlates with higher communicative complexity. Yet, there is still an inherent link between the occurrence of signals and other behaviours that allow the authors to determine the outcome of an interaction.

I do agree with the authors' conclusion that the three species vary in terms of the predictability of their signaling behaviour and the outcome of interactions. I just think the observed link between the two is not very surprising or informative, but rather inevitable.

Secondly, I am still not convinced that visual communication is more prevalent in situations with higher predation pressure. There are two reasons: relying on visual communication requires that the recipients, typically one's group members, are actually looking at the signaler when they produce the signal. The vocal-auditory channel in contrast, has a much higher potential to reach all recipients, even when visual communication in impaired. In addition, the idea that predators use acoustic signals to single out individuals and preferentially attack them, is poorly corroborated by data, especially for terrestrial predators. In contrast, there is ample evidence that prey species direct their calls at terrestrial predators (mobbing calls against snakes, antelope vigorously snorting against lions and leopards). See also this paper by Griesser (PMID 23941356).

---

## [Referee Report · Reviewer #2 (Public Review)]

This is a well-written manuscript about a strong comparative study of diversity of facial movements in three macaque species to test arguments about social complexity influencing communicative complexity.

---

## [Author Response]

The following is the authors’ response to the original reviews.

**Reviewer #1 (Public Review):**
This study investigates the context-specificity of facial expressions in three species of macaques to test predictions for the 'social complexity hypothesis for communicative complexity'. This hypothesis has garnered much attention in recent years. A proper test of this hypothesis requires clear definitions of 'communicative complexity' and 'social complexity'. Importantly, these two facets of a society must not be derived from the same data because otherwise, any link between the two would be trivial. For instance, if social complexity is derived from the types of interactions individuals have, and different types of signals accompany these interactions, we would not learn anything from a correlation between social and communicative complexity, as both stem from the same data.The authors of the present paper make a big step forward in operationalising communicative complexity. They used the Facial Action Coding System to code a large number of facial expressions in macaques. This system allows decomposing facial expressions into different action units, such as 'upper lid raiser', 'upper lip raiser' etc.; these units are closely linked to activating specific muscles or muscle groups. Based on these data, the authors calculated three measures derived from information theory: entropy, specificity and prediction error. These parts of the analysis will be useful for future studies.The three species of macaque varied in these three dimensions. In terms of entropy, there were differences with regard to context (and if there are these context-specific differences, then why pool the data?). Barbary and Tonkean macaques showed lower specificity than rhesus macaques. Regarding predicting context from the facial signals, a random forest classifier yielded the highest prediction values for rhesus monkeys. These results align with an earlier study by Preuschoft and van Schaik (2000), who found that less despotic species have greater variability in facial expressions and usage.Crucially, the three species under study are also known to vary in terms of their social tolerance. According to the highly influential framework proposed by Bernard Thierry, the members of the genus Macaca fall along a graded continuum from despotic (grade 1) to highly tolerant (grade 4). The three species chosen for the present study represent grade 1 (rhesus monkeys), grade 3 (Barbary macaques), and grade 4 (Tonkean macaques).The authors of the present paper define social complexity as equivalent to social tolerance - but how is social tolerance defined? Thierry used aggression and conflict resolution patterns to classify the different macaque species, with the steepness of the rank hierarchy and the degree of nepotism (kin bias) being essential. However, aggression and conflict resolution are accompanied by facial gestures. Thus, the authors are looking at two sides of the same coin when investigating the link between social complexity (as defined by the authors) and communicative complexity. Therefore, I am not convinced that this study makes a significant advance in testing the social complexity for communicative complexity hypothesis. A further weakness is that - despite the careful analysis - only three species were considered; thus, the effective sample size is very small.

Social tolerance in macaques is defined by various covarying traits, among which rates of counter-aggression and conflict resolution are only two of many included (see Thierry 2021 for a recent discussion and review). We do not deviate from Thierry’s definition of social tolerance. We simply highlight that the constellation of behavioral traits in the most tolerant macaque species results in a social environment where the outcome of social interactions is more uncertain (see introduction lines 102-114). As we argue throughout the paper, higher uncertainty can be used as a proxy for higher complexity and thus we conclude that the most tolerant macaque species have the highest social complexity. While most social behavior in macaques is accompanied by some facial behavior, we were careful to define social contexts only from the body language/behavior (e.g., lunge for aggression, grooming for affiliation) of the individuals involved and ignored the facial behavior used (see method lines 371-381). Therefore, the facial behavior of macaques (communication signals) was not used in defining either social tolerance (and by extension complexity) or the social context in which it was used. We feel like this appropriately minimizes any elements of circularity in the analysis of social and communicative complexity.

Regarding the effective sample size of three species, we agree that it is small, and it is a limitation of this study. However, the methodology we used is applicable to any species for which FACS is available (including other non-human primates, dogs, and horses), and therefore, we hope that other datasets will complement ours in the future. Nevertheless, we now acknowledge this limitation in the discussion (lines 314317).

**Reviewer #2 (Public Review):**
This is a well-written manuscript about a strong comparative study of diversity of facial movements in three macaque species to test arguments about social complexity influencing communicative complexity. My major criticism has to do with the lack of any reporting of inter-observer reliability statistics - see comment below. Reporting high levels of inter-observer reliability is crucial for making clear the authors have minimized chances of possible observer biases in a study like this, where it is not possible to code the data blind with regard to comparison group. My other comments and questions follow by line number:

We agree that inter-observer coding reliability is an important piece of information. We now report in more detail the inter-observer reliability tests that we conducted on lines 384-392.

38-40. Whereas I am an advocate of this hypothesis and have tested it myself, the authors should probably comment here, or later in the discussion, about the reverse argument - greater communicative complexity (driven by other selection pressures) could make more complicated social structures possible. This latter view was the one advocated by McComb & Semple in their foundational 2005 Biology Letters comparative study of relationships between vocal repertoire size and typical group size in non-human primate species.

It is true that an increase in communicative complexity could allow/drive an increase in social complexity. Unfortunately our data is correlational in nature and we cannot determine the direction of causality. We added such a statement to the discussion (lines 311-314).

72-84 and 95-96. In the paragraph here, the authors outline an argument about increasing uncertainty / entropy mapping on to increasing complexity in a system (social or communicative). In lines 95-96, though, they fall back on the standard argument about complex systems having intermediate levels of uncertainty (complete uncertainty roughly = random and complete certainty roughly = simple). Various authors have put forward what I think are useful ways of thinking about complexity in groups - from the perspective of an insider (i.e., a group member, where greater randomness is, in fact, greater complexity) vs from the perspective of an outside (i.e., a researcher trying to quantify the complexity of the system where is it relatively easy to explain a completely predictable or completely random system but harder to do so for an intermediately ordered or random system). This sort of argument (Andrew Whiten had an early paper that made this argument) might be worth raising here or later in the discussion? (I'm also curious where the authors sentiments lie for this question - they seem to touch on it in lines 285-287, but I think it's worth unpacking a little more here!)

In this study we used three measures of uncertainty (entropy, context specificity, and prediction error) to approximate complexity. However, maximum entropy or uncertainty would be achieved in a system that is completely random (and thus be considered simple). Therefore, the species with the highest entropy values, or unpredictability, could be interpreted as having a simpler communication system than a species with a moderately high entropy/unpredictability value. Our argument is that animal communication systems cannot possibly be random, otherwise they would not have evolved as signals. In systems where we know the highest entropy (or unpredictability) will not be due to randomness, as is the case with animal social interactions and communication, we can conclude that the system with the highest uncertainty is the most complex. We have now expanded upon this point in the discussion (lines 286-294). See also response to reviewer 1 below.

115-129. See also:Maestripieri, D. (2005). "Gestural communication in three species of macaques (*Macaca mulatta*, M. nemestrina, M. arctoides): use of signals in relation to dominance and social context." Gesture 5: 57-73.Maestripieri, D. and K. Wallen (1997). "Affiliative and submissive communication in rhesus macaques." Primates 38(2): 127-138.On that note, it is probably worth discussing in this paragraph and probably later in the discussion exactly how this study differs from these earlier studies of Maestripieri. I think the fact that machine learning approaches had the most difficulty assigning crested data to context is an important methodological advance for addressing these sorts of questions - there are probably other important differences between the authors' study here and these older publications that are worth bringing up.

Our study differs from these two studies in that the studies above classified facial behavior into discrete categories (e.g., bared-teeth, lip-smack), whereas we adopted a bottom-up approach and made no a priori assumptions about which movements are relevant. We broke down facial behavior down to their individual muscle movements (i.e., Action Units). Measuring facial behavior at the level of individual muscle movements allows for a more detailed and objective description of the complexity of facial behavior. This is a general point in advancing the study of facial behavior that is discussed in the introduction (lines 60-71) and discussion (lines 206-208). The reason we don’t draw a direct comparison with the studies above is because they had a slightly different focus. Our study was more focused on complexity of the (facial) communication system in general rather than comparing whether the different species use the same facial behavior in the same/different social contexts.

220-222. What is known about visual perception in these species? Recent arguments suggest that more socially complex species should have more sensitive perceptual processing abilities for other individuals' signals and cues (see Freeberg et al. 2019 Animal Behaviour). Are there any published empirical data to this effect, ideally from the visual domain but perhaps from any domain?

This is an interesting point. We are not aware of any studies showing differences in visual perceptions within the macaque genus. Both crested macaques and rhesus macaques are able to discriminate between individuals and facial expressions in match-to-sample tasks with comparable performances (Micheletta et al., 2015a, 2015b; Parr et al. 2008; Parr & Heinz, 2009). Similarly, several macaque species are sensitive to gaze shifts from conspecifics (Tomasello et al. 1998; Teufel et al. 2010; Micheletta & Waller, 2012).

274-277. I am not sure I follow this - could not different social and non-social contexts produce variation in different affective states such that "emotion"-based signals could be as flexible / uncertain as seemingly volitional / information-based / referential-like signals? This issue is probably too far away from the main points of this paper, but I suspect the authors' argument in this sentence is too simplified or overstated with regard to more affect-based signals.

Emotion-based signals could, in theory, also produce flexible signals and it is possible that some facial expressions reflect an emotional state. However, some previous studies have suggested that facial expressions are only used as a display of emotion, rather than such signals having evolved for a different function such as announcing future intentions. In our study we found that macaques used, in some cases, the same facial expressions (i.e. combination of Action Units) in at least two different social contexts that, presumably, differed in their emotional valence. Thus, it is unlikely that particular facial expressions are bound to a single emotion. We think that this is an important point to make even though it is slightly beyond the scope of our paper.

288 on. Given there are only three species in this study, the chances of one of the species being the 'most complex' in any measure is 0.33. Although I do not believe this argument I am making here, can the authors rule out the possibility that their findings related to crested macaques are all related to chance, statistically speaking?

We are not aware of a way to rule out this possibility. However, we believe that we are appropriately cautious throughout the paper and acknowledge that having only investigated three species is a limitation of this study in the discussion (lines 314-317, see also our response to reviewer 1 above).

329-330. The fact that only one male rhesus macaque was assessed here seems problematic, given the balance of sexes in the other two species. Can the authors comment more on this - are the gestures they are studying here identical across the sexes?

We agree it would have been preferable to collect data on more than one male rhesus macaque, but that was unfortunately not possible. We are not aware of any studies showing differences in the use of facial behavior between male and female rhesus macaques. If differences exist, most likely these would occur in a sexual/mating context. However, in our study we only considered affiliative (non-sexual), submissive, and aggressive contexts, where we have no a priori reason to believe that there are sex differences.

354-371. Inter-observer reliability statistics are required here - one of the authors who did not code the original data set, or a trained observer who is not an author, could easily code a subset of the video files to obtain inter-observer reliability data. This is important for ruling out potential unconscious observer biases in coding the data.

We agree this is an important piece of information. We now report in more detail the inter-observer reliability tests that we conducted on lines 384-392:

“An agreement rating of >0.7 was considered good [Ekman et al 2002] and was necessary for obtaining certification. To obtain a MaqFACS coding certification, AVR, CP, and PRC coded 23 video clips of rhesus macaques and the MaqFACScodes were compared to the data of other certified coders (https://animalfacs.com).

The mean agreement ratings obtained were 0.85, 0.73, 0.83 for AVR, CP, and PRC, respectively. In addition, AVR and CP coded 7 videos of Barbary macaques with a mean agreement rating of 0.79. AVR and PRC coded 10 videos of crested macaques with a mean agreement rating of 0.74.”

**Reviewer #1 (Recommendations For The Authors):**
Given the long debate on the concept of information exchange in animal communication, I would also recommend being more careful with the term 'exchanges of information' (line 271). Perhaps it's better to be agnostic in the context of this paper.

As suggested, we now changed the phrasing to focus on the behavior of the animals, rather than suggesting that information is being exchanged (lines 270-273),

Line 281: "This result confirms the assumption that facial behaviour in macaques is not used randomly": the authors are knocking down a straw man. Nobody who has ever studied animal communication would consider that signals occur randomly. Otherwise, they would not have evolved as signals.

Indeed, nobody claims that animal communication signals are used randomly. Although it may be taken for granted, we feel it is worthwhile to reiterate this point, given that we used relative entropy and prediction error as measures of complexity. For instance, maximum entropy or unpredictability would be achieved in a system that is completely random (and thus be considered simple). Therefore, the species with the highest entropy values, or lowest predictability, could be interpreted as having a simpler communication system than a species with a moderately high entropy value. But if we are working under the assumption that animal communication systems cannot possibly be random, then we can conclude that the species whose communication system has the highest entropy is in fact the most complex. We tried to make this justification clearer in the discussion (lines 285-294).

I did not follow why there is a higher reliance on facial signals when predation pressure is higher. Apart from the fact that the authors cannot address this question, they may want to reconsider this idea altogether.

We now expand on the logic of why predation pressure might affect the use of facial signals (see lines 308-309): “When predation pressure is higher, reliance on facial signals could be higher than, for example vocal signals, such as to not draw attention of predators to the signaller.”

Technical comments:One methodological issue that requires clarification is what the units of analysis are. The authors write that each row in their analysis denoted an observation time of 500 ms. How many rows did the authors assemble? The authors mention a sample size of > 3000 social interactions in the abstract. How did they define social interactions? And how many 'time windows' of 500 ms were obtained? Did they take one window per interaction or several? If several, then how was this move accounted for in the analysis? The reporting needs to be more accurate here. Most likely, the bootstrapping took care of biases in the data, but still, this information needs to be provided.

We have now added some additional information to the method section. Social interactions for each context had the following definitions: “Social context was labeled from the point of view of the signaler based on their general behavior and body language (but not the facial behavior itself), during or immediately following the facial behavior. An aggressive context was considered when the signaler lunged or leaned forward with the body or head, charged, chased, or physically hit the interaction partner. A submissive context was considered when the signaler leaned back with the body or head, moved away, or fled from the interaction partner. An affiliative context was considered when the signaler approached another individual without aggression (as defined previously) and remained in proximity, in relaxed body contact, or groomed either during or immediately after the facial behavior. In cases where the behavior of the signaler did not match our context definitions, or displayed behaviors belonging to multiple contexts, we labeled the social context as unclear. Social context was determined from the video itself and/or from the matching focal behavioral data, if available.” (lines 371-382). The total duration of all social interactions per social context, and thus the number of 500ms windows/rows, have been added to Table 1 (lines 395-397). There were several 500ms windows per social interaction. All 500ms time blocks per interaction were used in the statistical analyses in order to retain all the variation and complexity of the facial behavior (Action Unit combinations) used by the macaques (lines 403-405). Indeed the bootstrapping procedure was used to account for any biases in the data.

Overall, I would recommend providing more information on the actual behaviour of the animals. The paper is strong in handling highly derived indices representing the behaviour, but the reader learns little about the animals' behaviour. Thus, it would be great if statements about the entropy ratio were translated into what these measures represent in real life. For context specificity, this is clear, but for entropy, not so much.

A high entropy ratio essentially suggests that a species uses a high variety of unique facial behavior/signals and all signals in the repertoire are used roughly equally often (rather than one facial behavior being used 90% of the time and others rarely used). We have tried our best to better explain this point in the introduction (lines 75-81) and discussion (lines 215-222). Discussing exactly what these signals are and what they mean was beyond the scope of this paper.

Line 106: nepotism, not kinship

Changed as suggested (line 106).

Line 113: I would avoid statements about how a monkey society is perceived by its members.

We think that noting how individuals may perceive their social environment is worthwhile when defining social complexity, so have retained this point but changed the phrasing to be more speculative (lines 112-113).

Line 329: I was very surprised that only one male was represented in the data for rhesus monkeys. The authors try to wriggle their way out of this issue in the supplementary material ("Therefore, we have no a priori reason to expect an overall difference in the diversity and complexity of facial behaviour between the sexes"), but I think this is a major shortcoming of the analysis. They should ascertain whether there are no sex differences in the other two species regarding their variables of interest. They could then make a very cautious case for there being no sex differences in rhesus either. But of course, they would not know for sure.

As with our response to reviewer 2 above, we agree that it would have been preferable to collect data on more than one male rhesus macaque, but that was unfortunately not possible. We are not aware of any studies showing differences in the use of facial behavior between male and female rhesus macaques. If differences exist, most likely these would occur in a sexual/mating context. However, in our study we only considered affiliative (non-sexual), submissive, and aggressive contexts, where we have no a priori reason to believe that there are sex differences. Looking at sex differences in the use of facial behavior would be a worthwhile study on its own, but it is outside the scope of this paper.

This paper would make a stronger contribution if it focussed on the comparative analysis of facial expressions and removed the attempt of testing the social complexity for communicative complexity hypothesis.

A comparative analysis of the contextual use of specific facial movements is important. But this paper is focused on making a more general comparison of the communication style and complexity across species. The social complexity hypothesis for communicative complexity is one of the key theoretical frameworks for such an investigation and allows us to frame our study in a broader context. We contribute important data on 3 species with methods that can be replicated and extended to others species. Therefore, we believe that it is a worthy contribution to investigations of the evolution of complex communication.

REFERENCES

Micheletta, J., J. Whitehouse, L.A. Parr, and B.M. Waller. ‘Facial Expression Recognition in Crested Macaques (Macaca nigra)’. Animal Cognition 18 (2015): 985–90. https://doi.org/10/f7fvnh.

Micheletta, Jérôme, Jamie Whitehouse, Lisa A. Parr, Paul Marshman, Antje Engelhardt, and Bridget M. Waller. ‘Familiar and Unfamiliar Face Recognition in Crested Macaques (Macaca nigra)’. Royal Society Open Science 2 (2015): 150109. https://doi.org/10/ggx9k9.

Parr, L. A., and M. Heintz. ‘Facial Expression Recognition in Rhesus Monkeys, *Macaca mulatta*’. Animal Behaviour 77 (2009): 1507–13. https://doi.org/10/bbsp5n.

Parr, L.A., M. Heintz, and G. Pradhan. ‘Rhesus Monkeys (*Macaca mulatta*) Lack Expertise in Face Processing’. Journal of Comparative Psychology 122 (2008): 390–402.https://doi.org/10/d7w6bv.

Micheletta, J., and B.M. Waller. ‘Friendship Affects Gaze Following in a Tolerant Species of Macaque, Macaca nigra’. Animal Behaviour 83 (2012): 459–67. https://doi.org/10/c4f8n2.

Thierry B. Where do we stand with the covariation framework in primate societies? Am. J. Biol. Anthropol. 128 (2021): 5–25. https://doi.org/10.1002/ajpa.24441

Tomasello, M., J. Call, and B. Hare. ‘Five Primate Species Follow the Visual Gaze of Conspecifics’. Animal Behaviour 55 (1998): 1063–69. https://doi.org/10/bmq7xh.

Teufel, C., A. Gutmann, R. Pirow, and J. Fischer. ‘Facial Expressions Modulate the Ontogenetic Trajectory of Gaze-Following among Monkeys’. Developmental Science 13 (2010): 913–22. https://doi.org/10/b6j5r7.